# Comparative analysis of tardigrade locomotion across life stage, species, and disulfiram treatment

**Emma M. Anderson, Sierra G. Houck, Claire L. Conklin, Katrina L. Tucci, Joseph D. Rodas📷, Kate E. Mori, Loriann J. Armstrong, Virginia B. Illingworth, Te-Wen Lo📷, Ian G. Woods📷***

Department of Biology, Ithaca College, Ithaca, New York, United States of America

* iwoods@ithaca.edu

**Data Availability Statement:** All sequence files are available from the GenBank database, https://www.ncbi.nlm.nih.gov/nucleotide (accession number(s) PP852068-PP852090) All code and datasets

## Abstract

Animal locomotion requires coordination between the central and peripheral nervous systems, between sensory inputs and motor outputs, and between nerves and muscles. Analysis of locomotion thus provides a comprehensive and sensitive readout of nervous system function and dysfunction. Tardigrades, the smallest known walking animals, coordinate movement of their eight legs with a relatively simple nervous system, and are a promising model for neuronal control of limb-driven locomotion. Here, we developed open-source tools for automated tracking of tardigrade locomotion in an unconstrained two-dimensional environment, for measuring multiple parameters of individual leg movements, and for quantifying interleg coordination. We used these tools to analyze >13,000 complete strides in >100 tardigrades, and identified preferred walking speeds and distinct step coordination patterns associated with those speeds. In addition, the rear legs of tardigrades, although they have distinct anatomy and step kinematics, were nonetheless incorporated into overall patterns of interleg coordination. Finally, comparisons of tardigrade locomotion across lifespan, between species, and upon disulfiram treatment suggested that neuronal regulation of high-level aspects of walking (e.g. speed, turns, walking bout initiation) operate independently from circuits controlling individual leg movements and interleg coordination.

## Introduction

Movement is a defining feature of animals, and is essential for obtaining food, finding suitable mates, locating favorable conditions in shifting environments, and escaping predation [1]. As movement is highly optimized and a primary determinant of morphology and physiology [2, 3], quantitative analysis of movement enables insight into key features of an animal's form and function. Movement requires coordination between the peripheral and central nervous systems, signaling between nerves and muscles, and response to external and internal cues [1, 4, 5], and is a sensitive indicator of nervous system function [6, 7]. For example, walking ability predicts healthspan and life expectancy in elderly humans, and abnormal gait can reveal neurological conditions before clinical diagnoses [8, 9]. Similarly, movement analysis can uncover

needed to produce the figures and tables in this study are available at GitHub http://www.github.com/enwudz/gait All videos of walking tardigrades underlying this study are available at Open Science Framework http://doi.org/10.17605/OSF.IO/XBKA2 All protocols used for tardigrade handling, preparation for videography, video acquisition, path tracking, step timing quantification, and analysis, along with descriptions of Python scripts used in the analysis pipeline, are available at protocols.io http://dx.doi.org/10.17504/protocols.io.kxygxy8nwl8j/v1.

**Funding:** National Science Foundation Louis Stokes Alliances for Minority Participation (award number 2110082) William T. Grant Foundation (Grant ID #ODF-203161) Ithaca College Collegiate Science and Technology Entry Program Ithaca College Humanities and Sciences Summer Scholars Program.

**Competing interests:** The authors have declared that no competing interests exist.

**Abbreviations:** CCS, Coordination Consistency Score; CSS, Coordination Strength Score; ICP, Interleg Coordination Pattern.

the etiology of many pediatric disorders [10], and help diagnose neurological dysfunction in animals [11].

The complex neurobiology of vertebrate appendage-based locomotion makes a circuit-level understanding a major challenge. Simpler or smaller animals could provide more tractable models, but most microscopic animals move via axial propulsion with undulating or peristaltic movements rather than via alternating limbs. Named for their slow, stepping movements, tardigrades are a notable exception. Tardigrades generate appendicular locomotion with a relatively simple nervous system: their bodies consist of < 2000 cells in species where cell number has been measured, and only a few hundred of these cells are neurons [12–15]. Tools for genetic manipulation are actively being developed for tardigrades [16] making them an attractive system for investigating the genetics and neuromuscular biology underlying appendicular locomotion.

To establish tardigrades as an accessible model for locomotor function and dysfunction, we developed open-source and easily-implemented tools to quantify exploratory locomotion, single-leg stepping, and interleg coordination. To validate these tools, we confirmed key observations of a previous analysis of tardigrade walking [17]. In total, we tracked 103 *Hypsibius exemplaris* individuals over extended times and distances, thereby producing a comprehensive dataset of intraspecies variation in locomotor behavior. We extend knowledge of tardigrade locomotion via automated quantification of exploratory walking over large temporal and spatial scales, by documenting preferences in interleg coordination at different speeds, by measuring coordination between specialized appendages, and by analyzing differences in locomotion across life stages, species, and pharmacological treatments.

## Methods

### Tardigrade culture

*Hypsibius exemplaris* [18] were obtained commercially (Sciento, Manchester, England, strain Z151) and maintained in the laboratory in 50 mm Petri dishes coated with 2% agar and filled with 5.0 mL Poland Spring water. Cultures were refreshed every 1–2 weeks by changing the water and adding 5 drops of fresh algae for food (*Chlorococcum*, initially obtained from Carolina Biological Supply, Burlington, NC, USA, catalog 152091, and cultured following procedures from the supplier). Stocks were regenerated every 2–4 weeks by transferring 10–15 adult tardigrades to new agar-coated dishes and adding fresh algae.

### Videos of tardigrade locomotion

To quantify tardigrade locomotion, we transferred individuals to 35 mm Petri dishes freshly coated with 2% agarose, and filled each dish with 1 mL of Poland Spring water after transfer. Optimal agarose concentration was determined by testing a range between 1% and 3%. At high concentrations, tardigrades frequently lost their grip on the substrate and were unable to walk, whereas lower substrate stiffness is associated with slow movement and changes in stepping patterns [17]. Tardigrades were filmed at 33 Hz at 1024x768 resolution (AmScope MD310B-BS) under a 10X objective on an inverted microscope. During filming (AmScopeAmLite v3.26.2023), we minimized stage movement, thus maximizing the time during which the background remained stationary. After filming, we trimmed each video into clips with motionless background.

### Automated tracking and quantification of movement

We developed Python scripts to detect tardigrades and measure size, speed, and direction of travel (www.github.com/enwudz/gait). The outline of the tardigrade in each video frame was

used to determine area, length, and width, and center of mass coordinates. For a subset of tardigrades, these measurements were validated via additional image analysis software (e.g. FIJI, https://imagej.net/software/fiji/). Instantaneous speed and direction of travel was determined by comparing coordinates between video frames. A 'stop' was recorded when a tardigrade moved $\leq 0.03$ body lengths in 0.3 seconds, and a 'turn' occurred when a tardigrade changed direction by $\geq 28°$ during 0.3 seconds. These thresholds were determined by iteratively optimizing the scripts until they matched qualitative calls of stops and turns in pilot videos. Forward motion without stops and turns was defined as a sustained walking bout. Tracking was validated for all individuals by plotting the complete path of travel, and for a subset of individuals by creating a new movie showing the walking tardigrade, its tracked path in real time, and the automated calls of stops and turns. Details of the tracking scripts and instructions for their use are available at protocols.io (http://dx.doi.org/10.17504/protocols.io.kxygxy8nwl8j/v1).

## Step kinematics

Sustained walking bouts of 3 seconds or longer were used to collect timing of swings and stances for individual legs. For each tardigrade, we selected 1–3 sustained walking bouts to analyze a total of at least 10 seconds of walking. If multiple candidate bouts were available, we prioritized individual bouts of ~8 seconds, corresponding to the time it takes for a typical tardigrade to travel across the span of a video frame.

We developed Python scripts to record step and swing timing semi-automatedly for each video clip. For each leg, we stepped through a sustained walking bout frame by frame, and recorded step timing via specific keystrokes as the claw contacted or released contact with the agarose surface. From timing of steps and swings for each complete stride (foot down to foot down), our scripts calculated stance and swing duration, duty factor (proportion of each stride in stance phase), step period (elapsed time between stance initiations), and swing-swing offset timing between pairs of legs. When combined with the path tracking data above, step and swing times were used to calculate stride length (distance in floor-fixed coordinates between two consecutive stance onsets), stride speed, and change in direction per stride.

## Interleg coordination

As in previous studies, we assigned a momentary coordination pattern to each video frame according to identities of legs that were in swing phase [17, 19–21]. For initial interleg coordination analyses, the first three leg pairs ('lateral legs') were considered independently from the rear legs, as in a previous study of tardigrade locomotion [17]. Frames with all lateral legs in stance phase were classified as 'stand', and periods with one lateral leg in swing phase were classified as 'pentapod.' We subdivided tetrapod coordination patterns (two legs in swing phase) into three categories: (1) tetrapod canonical, when the two swinging legs were diagonally oriented (i.e. contralateral to each other in adjacent segments), (2) tetrapod gallop, when the two swinging legs were within the same body segment, and (3) tetrapod other, when the two swinging legs did not match the canonical or gallop pattern. Similarly, we subdivided tripod patterns into two categories: (1) tripod canonical, when the three swinging legs were diagonally oriented, contralaterally in adjacent segments, and (2) tripod other, when the three swinging legs did not match the canonical tripod pattern. Finally, we divided rear leg stepping patterns into three groups: (1) stand (both rear legs in contact with the substrate), (2) step (one leg in swing phase), and (3) hop (both legs in swing phase).

We measured consistency of coordination patterns over successive strides via a coordination consistency score (CCS). To calculate CCS, we identified the most prevalent canonical tetrapod ICP for each leg within each recorded bout of walking, and counted the number of

strides that contained that specific ICP. For example, if the first left leg (L1) had 10 complete strides within a walking bout, and in 8 of these strides the second right leg (R2) was also in swing phase, then our totals after considering L1 would be 8 strides within a particular ICP, and 10 strides total. We repeated this procedure for each lateral leg, and added these values to running totals of per-leg ICP strides and total strides. After all lateral legs were completed, we divided the total per-leg ICP strides by the total strides, and assigned each tardigrade a single score ranging from 0 to 1, where CCS = 1 would mean that each leg exhibited a specific ICP that was repeated within all of its strides.

As an additional measure of interleg coordination, we calculated the coordination strength score (CSS) for each bout of canonical tetrapod or canonical tripod walking [22]. CSS was defined as the ratio of time during which a set of legs maintained a particular pattern versus the elapsed time between first swing onset and the last stance onset of all legs involved in that pattern. For example, a canonical tetrapod bout would be called in video frames where L1 and R2 were simultaneously in swing phase. If L1 initiated swing 0.5 seconds before R2, both legs were in swing phase for 1 second, and L1 initiated stance 0.5 seconds before R2, then the CSS for this bout would be 0.5 (1 second where both L1 and R2 were swing phase / 2 seconds between swing onset of L1 and stance onset of R2). A CSS = 1 would mean that swing and stance onsets were simultaneous for all legs participating in the ICP. Each tardigrade was assigned a tetrapod and tripod CSS score based on the average CSS across all bouts of canonical tetrapod or tripod coordination.

## Isolation of wild *Ramazzottius* and 18S rDNA sequencing

Moss and lichen were collected from trees on the Ithaca College campus, and soaked overnight in Poland Spring water. Candidate *Ramazzottius* individuals were identified via their characteristic claw morphology and striped pigmentation. After videography, *Ramazzottius* individuals were isolated for 18S rDNA sequencing. Identities of a subset of the lab *Hypsibius exemplaris* individuals and wild-caught *Ramazzottius* individuals were confirmed by sequencing of 18S ribosomal DNA and alignment via BLAST with nucleotide sequences in GenBank (https://www.ncbi.nlm.nih.gov/nucleotide/). Tardigrade gDNA was prepared via the Hot-SHOT technique [23], and amplicons for all sequenced individuals were obtained via Polymerase Chain Reactions using the primers: F566: 5'-CAGCAGCCGCGGTAATTCC-3', R1200: 5'-CCCGTGTTGAGTCAAATTAAGC-3' [24]. All lab tardigrade sequences (N = 14) matched GenBank accession number MG800327 (*Hypsibius exemplaris* strain Sciento Z151) via BLAST.

## Isolation of juvenile tardigrades

*Hypsibius exemplaris* adults deposit eggs in their discarded cuticle (exuvium) during molting. To isolate freshly-hatched juvenile tardigrades, we transferred individual egg-containing exuviae to empty agar-coated Petri dishes, and photographed each exuvium at the same time on each subsequent day. Walking tardigrades were prepared for video tracking as soon as they were found, thereby ensuring that fewer than 24 hours had elapsed after hatching. When possible, we filmed juvenile tardigrades immediately after they emerged from their exuvium.

## Disulfiram treatment

Dosage for disulfiram was determined by testing a range based on exposure estimates from previous studies [25, 26], and selecting the dose within that range that resulted in locomotor disruption (e.g. slower movement), but intact walking ability. For each disulfiram experiment, we selected 20–24 large tardigrades with bright green gut contents, and distributed them evenly between two Petri dishes coated with 2% agarose. We avoided tardigrades that appeared

brown rather than green, as these brown tardigrades typically contain large eggs and, when treated with disulfiram, often become trapped within their own exuviae after egg expulsion. We prepared 5 mL of control and 1.1 μM disulfiram solutions of Poland Spring water and 1% DMSO, and added these solutions to the two dishes. We added 5 drops of concentrated *Chloroccum* to each dish, and incubated the two dishes within a single box in the dark, overnight at room temperature. After 24 hours of incubation, we screened each dish for walking tardigrades, and prepared them for videography as above.

## Statistics

Regression fit lines, histogram density distributions, and probability density functions were generated via the Python package seaborn [27]. Data for pair-wise comparisons were evaluated for normal distributions via the Shapiro-Wilk test, and Levene's test was conducted to assess equal variance. A two-tailed t-test was used in pairwise comparisons of normally distributed data, while the Mann—Whitney rank sum test was used to compare data that was not normally distributed. To correct for multiple hypothesis testing, the False Discovery Rate was controlled at 5% via the Benjamini-Yekutieli procedure. Statistical tests were conducted in the Python scipy.stats module [28]. To compare timing within particular coordination patterns, proportions were transformed into log ratios via the center log ratio method in the Python composition_stats module.

## Results

### Automated tracking of exploratory locomotion

We filmed 103 single *Hypsibius exemplaris* individuals roaming freely on agarose-coated Petri dishes, and developed methods for automated quantification of tardigrade locomotion over extended distances and times (Fig 1). Tardigrades were free to start and stop walking, to turn, and to engage in extended bouts of locomotion in any direction. We captured video for an average of 139 (138.61 ± 23.45, mean ± s.d.) seconds, and trimmed videos into clips with stationary backgrounds (totaling 97.26 ± 18.67 seconds, Fig 1B, S1 Movie). Exploratory walking was tracked over 22 body lengths on average (22.25 ± 7.02 body lengths, S1 Table). All videos analyzed in this study are available at Open Science Framework (https://osf.io/xbka2/).

Automated measurements of body size and location were calculated in each video frame (Fig 1A). Mean tardigrade length (251.53 ± 32.25 μm) was within the range described for *Hypsibius exemplaris* (112–293 μm) [18], and lengths were strongly correlated with widths (Fig 1C, Spearman's ρ = 0.91; p<0.001, n = 103). Speed was calculated in physical distance (μm s$^{-1}$) and normalized to length (body lengths s$^{-1}$), and stops and turns were automatically called by our scripts. Stops were often followed by turns: tardigrades ceased walking, compressed posteriorly, and oriented toward a new direction (Fig 1D).

Tardigrades spent an average of 89% (88.68 ± 10.93) of time in sustained walking (without turns or stops, Fig 1E), and moved an average of 0.23 (0.23 ± 0.08) body lengths per second (S1 Table). Size was correlated with speed in absolute units (S1A and S1B Fig), but not when speed was normalized to body length (Fig 1H) or area (S1C Fig). Tardigrades with a high width: length ratio moved more slowly (Spearman's ρ = -0.37, p<0.001, n = 103, Fig 1I).

### Confirmation of step kinematics and interleg coordination

From the videos of exploratory locomotion, we quantified step kinematics and interleg coordination during bouts of sustained walking (Fig 2A–2G). Because details of video capture and walking substrate differed from a previous study that measured these parameters in

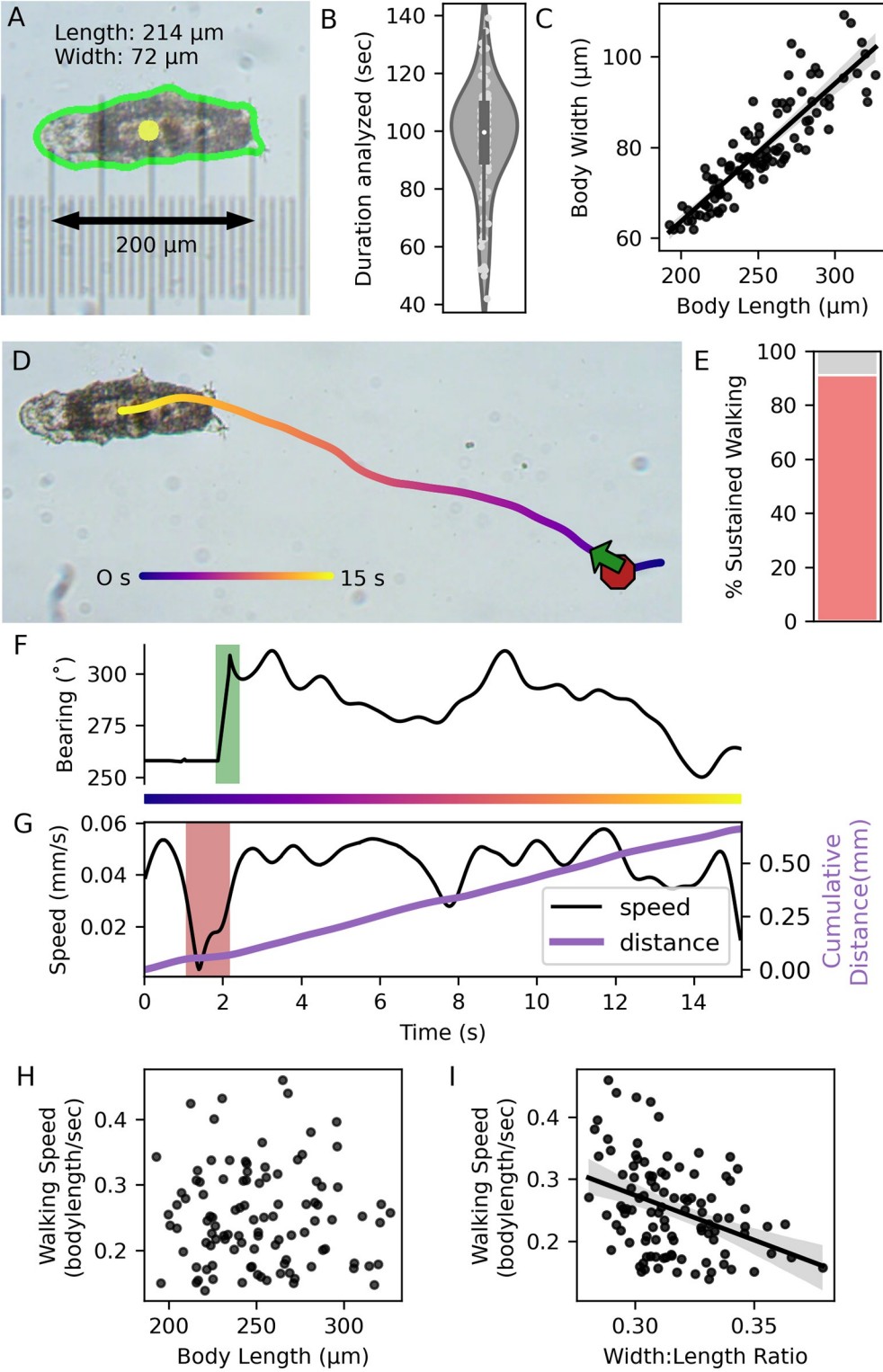

**Fig 1. Tracking and quantification of locomotion.** A. Single video frame. Tardigrade outline (green) was automatically detected, and used to calculate length, width, and center of mass (yellow). A micrometer (at same magnification and resolution) was superimposed to verify measurements. B. Video durations analyzed in 103 tardigrades. C. Body width was strongly correlated to length (Spearman's ρ = 0.905, p<0.001, n = 103). D. Last frame of a ~15 second video, showing the path traveled, including a stop (red octagon) and a turn (green arrow). E.

Proportion of sustained walking during the video from (D). F. Direction traveled during the video from (D); 0˚ = up, green rectangle = turn. G. Speed and distance traveled during the video from (D). Red rectangle = stop. H. Normalized speed was not correlated with tardigrade length (Spearman's $\rho$ = -0.003, p = 0.98, n = 103). I. Speed was inversely proportional to the width:length ratio (Spearman's $\rho$ = -0.37, p<0.001, n = 103). Least squares regression fit lines and 95% confidence intervals (C, I) were calculated in the Python package seaborn.

tardigrades [17], and because our analysis code and procedures were developed independently, we first compared our results to the previous study.

We recorded stance and swing initiation (Fig 2A and 2B), and analyzed ~128 (127.8 ± 41.5) complete strides over 14 (14.4 ± 3.2) seconds per individual (S1 Table). Rear legs have distinct musculature, innervation, body orientation, and function [29], so we considered the first three leg pairs ('lateral legs') separately, as in the previous study [17]. We measured 9471 strides in lateral legs, and 3692 in rear legs. From stance and swing durations (Fig 2C), we calculated duty factors (proportion of a stride instance, Fig 2D), step periods (timing between stance onsets), and distance traveled for each stride (Fig 2E).

As shown previously [17], tardigrades increased speed by decreasing stance duration (Fig 3A), whereas swing duration decreased only slightly with increased speed (Fig 3B). Duty factors were inversely correlated with speed for all legs (first leg pair: Spearman's $\rho$ = -0.8; p<0.001; fourth leg pair: Spearman's $\rho$ = -0.65, p<0.001, n = 103 individuals, Fig 3C). Compared to lateral legs, rear legs showed shorter stances and longer swings, leading to lower duty factors (Figs 2C and 3C), but similar step periods (Fig 3D).

To quantify ipsilateral interleg coordination, we measured offsets between swing initiations of neighboring legs along each side; these values were normalized to the step period of the first leg in a paired comparison, and expressed as phase offsets ($\phi_I$) ranging from 0 to 1 (Fig 2F). Phase offsets and duty factors determine which legs are simultaneously in swing phase, and combinations of swinging legs define **i**nterleg **c**oordination **p**atterns (ICPs) when sustained over time (S2A Fig). Following the procedure of the previous study [17], we categorized coordination of lateral legs into hexapod ICPs, including pentapod (one swinging leg), tetrapod, and tripod patterns. Combinations with more than three swinging legs are rare, presumably because they are unstable [30]. In hexapods, $\phi_I$ = 1/3 and 2/3 lead to tetrapod ICPs, while $\phi_I$ = 0.5 generates a tripod ICP (S2B—S2E Fig). As previously observed [17], the distribution of $\phi_I$ peaked at 1/3, (Fig 4A and 4D); this distribution was consistent across lateral legs (S3A Fig).

In a walking ICP (S2 Fig), neighboring ipsilateral legs rarely swing simultaneously. This pattern can be modeled by two simple rules: (1) swing onsets are suppressed while a neighboring leg is in swing phase, and (2) the likelihood of swing onset increases after the stance onset of a neighboring leg [30–33]. As in the previous study [17], this pattern of suppression and release is clearly evident in our dataset (S4A, S4B, S4F and S4G Fig).

To measure coordination between contralateral legs, we calculated the contralateral phase offset ($\phi_C$) by comparing swing onsets within each body segment. For all legs, $\phi_C$ was centered around 0.5, indicating predominantly antiphase movement within contralateral leg pairs (Fig 4B and 4D). The distribution of $\phi_C$ of lateral legs was qualitatively broader than that of $\phi_I$ (Fig 4A and 4B), suggesting that contralateral coordination is more flexible than ipsilateral coordination, in agreement with previous analyses [17, 32, 34]. Accordingly, coupling between leg movements is less pronounced for contralateral leg pairs than for ipsilateral pairs (S4A and S4F Fig).

In our dataset, the greatest density of steps occurred at the intersection of $\phi_I$ = 1/3 and $\phi_I$ = 2/3, typical of a tetrapod ICP with asynchronous swing onsets of the 2 swinging legs (Fig 4D and 4E, S2D Fig). As in previous studies [20, 21], we assigned a momentary hexapod ICP for

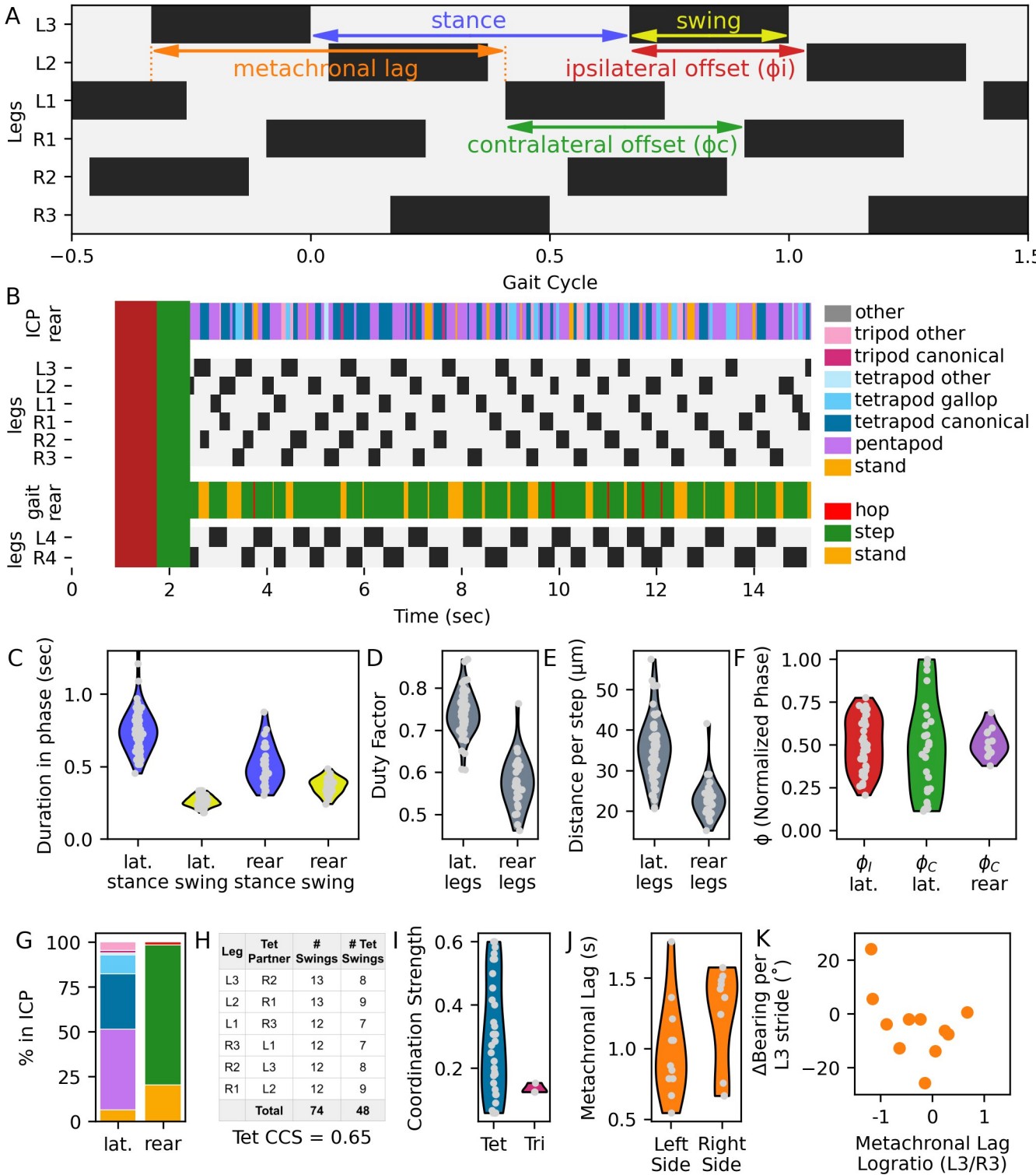

**Fig 2. Quantifying step kinematics and interleg coordination in a single video clip.** A. Example lateral leg stepping pattern. Stance (gray) and swing (black) duration were measured for each leg, and offset timing was calculated by comparing swing initiations between neighboring ipsilateral legs or contralateral legs within a segment. Metachronal lag = elapsed time between swing onsets of a third leg and a first leg on one side, with an intervening swing in the second leg. B. Stance and swing timing and interleg coordination patterns (ICPs) for the video in Fig 1D. Red rectangle = stop; green = turn. Swing and stance onsets were recorded during sustained walking. Within each frame, the combination of legs in swing phase determined the ICP, with lateral legs (first three pairs, top) and rear legs (fourth pair, bottom) considered independently. L1 = anteriormost left leg; other legs are labeled similarly. C. Stance and swing duration for all steps.

D. Duty factor—the proportion of each stride spent in stance—for all steps. E. Distance per step. F. Phase offsets ($\phi_I$ = ipsilateral, $\phi_C$ = contralateral). G. Percentage of the clip spent in indicated ICPs for lateral and rear legs, with color codes as in (B). H. Calculation of coordination consistency score (CCS) for canonical tetrapod ICP within this clip. I. Coordination strength scores (CSS) for canonical tetrapod or tripod ICP bouts within this clip. J. Metachronal lag on the left vs. the right sides. K. Symmetry of metachronal lag plotted against change in direction during each stride of the left third leg.

every video frame, based on the identities of swinging lateral legs (Fig 2B). Tetrapod patterns were most common (48%, with 37.5% as canonical tetrapod), followed by pentapod (35.8%), and tripod (8.3%, with 2.9% as canonical tripod) patterns (Fig 4G, S1 Table). Our measurements of step kinematic parameters and interleg coordination for lateral legs thus closely matched those of the previous study [17], indicating that tardigrades walk similarly on different substrates, and that the different experimental approaches converged on similar behavioral observations.

## Quantifying coordination

Having validated our approach and tools, we proceeded with new analyses. While a frame-by-frame snapshot of legs in swing phase provides an estimate of overall interleg coordination (Fig 2B and 2G), it is important to note that coordination involves a temporal component that is not directly measurable in a snapshot. To assess maintenance of ICPs through time, we developed a coordination consistency score (CCS), which measures the proportion of strides for each lateral leg that contain a specific coordination pattern throughout the measured bout of sustained walking. Each tardigrade received a CCS score ranging from 0 to 1; CCS = 1 means lateral legs participated in a specific ICP that repeated during each of their strides (Fig 2H). Average tetrapod CCS (0.65 ± 0.14, Fig 4F, S1 Table) was correlated with walking speed

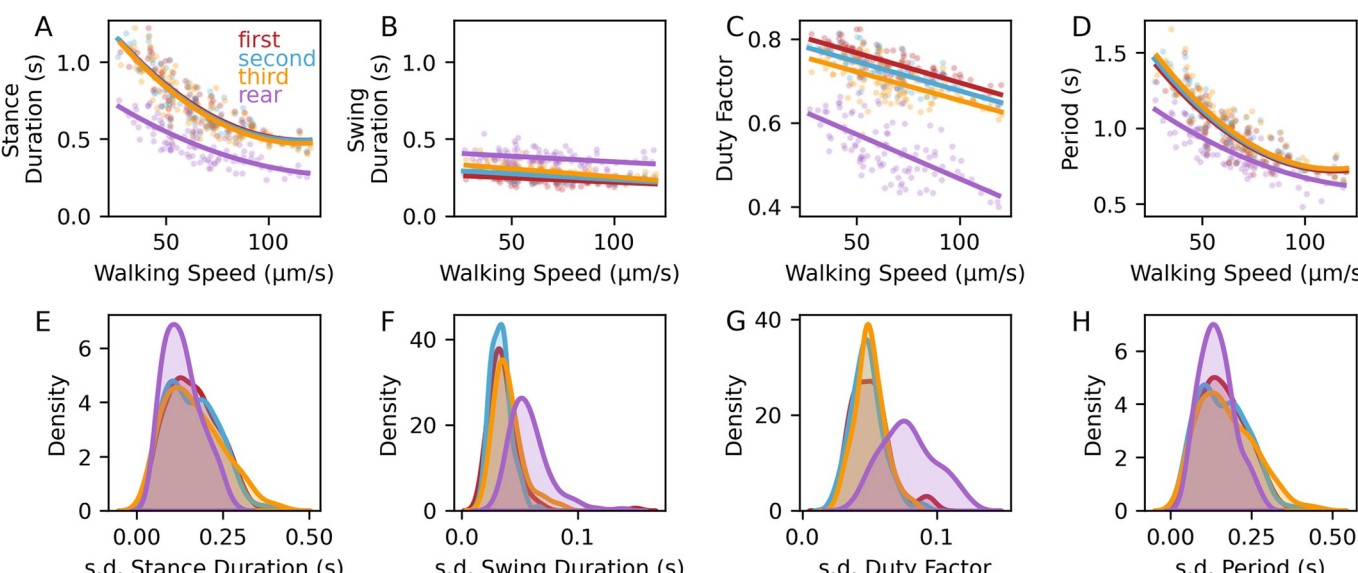

**Fig 3. Distinct step kinematic parameters in lateral and rear legs.** A. Average stance duration decreased with increased walking speed, and was shorter in rear legs. B. Swing duration remained roughly constant with increasing speed, and was longer in rear legs. C-D. Duty factor and step period decrease with increased speed. E-H. Within-animal standard deviations for stance (E) and swing (F) durations, duty factor (G), and period (H). Regression fits (A-D) and probability densities (E-H) were calculated via the Python package seaborn. N = 103 tardigrades for all panels.

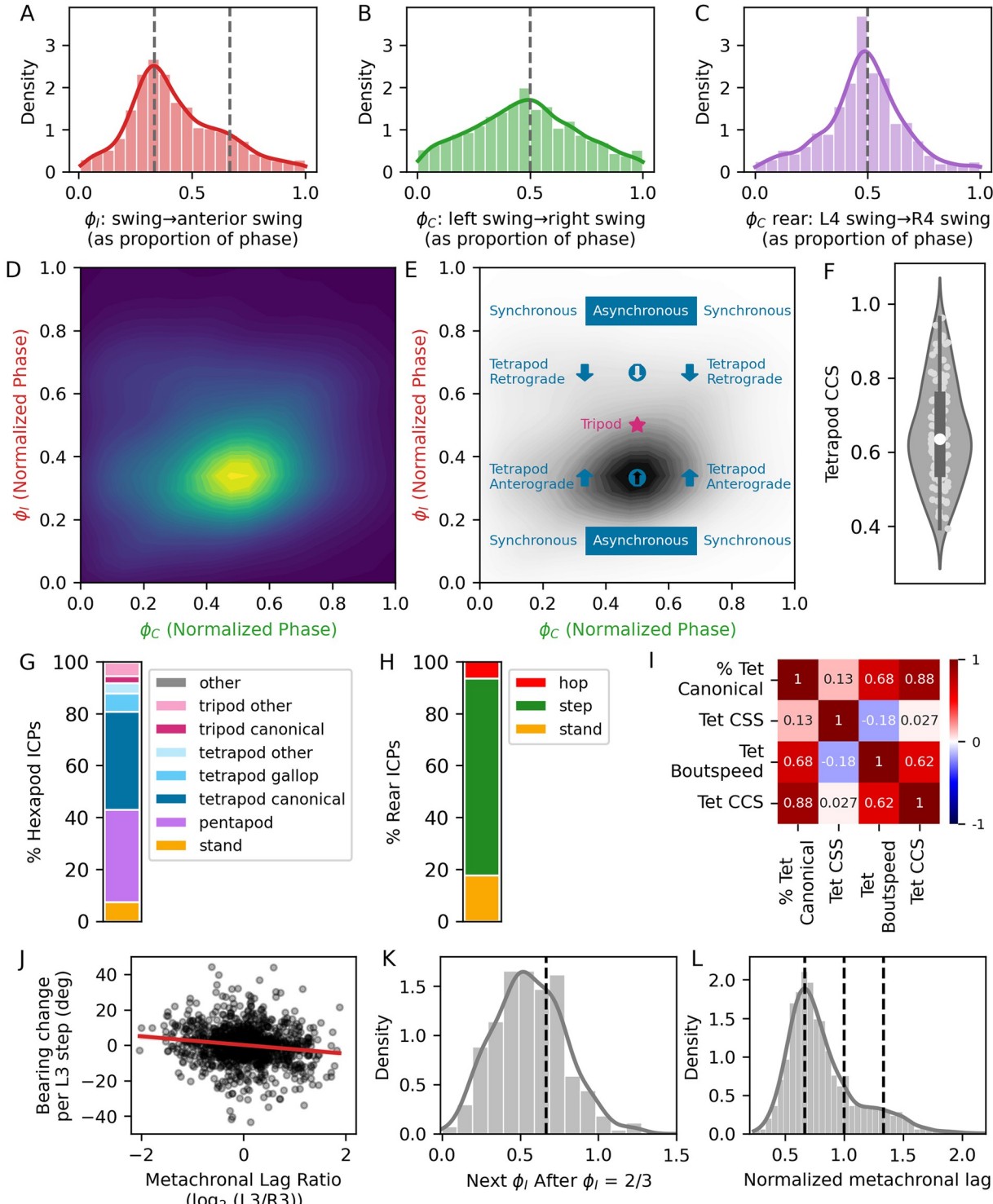

**Fig 4. Interleg coordination in 103 adult *Hypsibius exemplaris* individuals.** A. Distribution of ipsilateral phase offsets ($\phi_I$): swing initiation timing between lateral legs (second or third pair) and neighboring ipsilateral anterior legs (first or second), expressed as a fraction of the posterior leg step period. Dashed lines at 1/3 and 2/3 indicate offsets underlying canonical tetrapod gait, n = 5175 strides. B. Distribution of contralateral phase offsets ($\phi_C$): timing of swing initiation between left lateral legs and contralateral (right) legs, expressed as a fraction of the left leg step period, n = 3809 strides. C. $\phi_C$, rear: swing initiation timing between the left and right rear legs, expressed as a fraction of the left rear leg step period. n = 1530 strides. D. Probability density plot of $\phi_C$ vs. $\phi_I$ for second and third pair legs, n = 4414 strides. E. Diagram showing relationship between $\phi_I$, $\phi_C$, and canonical tetrapod or tripod ICPs, and observed distribution of $\phi_C$ vs. $\phi_I$ for for second and third pair legs. The

highest density occurs at a canonical tetrapod pattern, with apparent anterograde wave of swings, and asynchronous swing initiations between the two legs participating in the pattern at each instant. F. Tetrapod Coordination Consistency Score (CCS) for 103 tardigrades (light points). Box represents interquartile range, and white circle denotes the median. G,H. Percentage of video frames spent in indicated hexapod (G) or rear leg (H) ICPs. I. Pearson correlation coefficients for indicated paired comparisons (n = 103 tardigrades). J. Log$_2$ ratio of metachronal lag of (L3/R3) plotted against direction change for each L3 step (n = 1299 strides). Pearson ρ = -0.15, p < 0.001. Least squares regression fit line was generated via the Python package seaborn. K. Distribution of $\phi_i$ in third legs after a stride where $\phi_i$ was ~2/3. The dashed line indicates $\phi_i$ = 2/3. n = 277 strides. L. Distribution of metachronal lag for all third legs. Dashed lines indicate values of 2/3, 1, and 4/3. n = 2793 strides.

(Fig 4I, Pearson ρ = 0.62; p<0.001, n = 103), and with percentage of strides spent in any canonical tetrapod ICP (Pearson ρ = 0.88; p<0.001, n = 103).

Interleg coordination can also be described via a coordination strength score (CSS, [22]). When CSS = 1, all legs participating in an ICP bout have simultaneous swing and stance onsets. We calculated CSS during canonical tetrapod and canonical tripod bouts for each tardigrade (S1 Table). Average per-individual tetrapod CSS (0.39 ± 0.06) and tripod CSS (0.16 ± 0.07) were not correlated with walking speed (Fig 4I). These CSS scores indicate that while legs involved in ICP bouts differed in timing of stance and swing onsets, a substantial proportion of each lateral leg stride was, on average, spent in a canonical tetrapod ICP (Fig 2I).

## Integrating step kinematics and exploratory locomotion

The size and unique features of our dataset enabled additional analyses not previously explored in tardigrades. For example, simultaneous recording of exploratory behavior (e.g. turning) and step kinematics enables exploration of potential relationships between these parameters. To quantify left-right symmetry of ipsilateral leg coordination (Fig 2J and 2K), we measured the metachronal lag on each side and calculated the absolute value of the log$_2$ ratio (0 = symmetric). Over the ~14 seconds analyzed per tardigrade, this ratio was 0.47 (0.47 ± 0.2), indicating a difference of about 38% in metachronal lag between the two sides (S1 Table). This asymmetry could be generated by turns or by stochastic differences in coordination. To test the relationship between metachronal lag and turning (Fig 2K), we plotted the change in direction in each third leg stride against the log$_2$ ratio of (left metachronal lag/right metachronal lag) and found that longer metachronal lag on the left (i.e. log$_2$(L3/R3) > 0) was weakly correlated with turning to the left (Pearson's ρ = -0.15, p<0.001, n = 1299 strides, Fig 4J).

## Leg specialization and coordination

Compared to lateral legs, rear legs show differences in innervation, musculature, orientation of the legs with respect to the body, increased ipsilateral distance between rear and lateral legs, and may have a primarily grasping (rear) vs. walking (lateral) function [17]. However, rear legs can participate in walking in *Hypsibius exemplaris*, but have distinct stance and swing timing compared to the lateral legs (Figs 2 and 3). In addition, rear legs show distinct contralateral coordination: $\phi_C$ for rear legs was tightly clustered at 0.5 (Fig 4C), suggesting stronger anti-phase coordination than in lateral legs. Accordingly, coupling between rear leg swing duration and contralateral leg swings is more pronounced than in lateral legs (S4D and S4E Fig). We therefore categorized rear ICPs separately, and found that steps were most common (75.8%), followed by stand (17.6%) and hop (6.5%, Figs 4H and 5E, S1 Table).

Nevertheless, when rear legs participate in walking, their step period was similar to that of lateral legs, especially at higher speeds (Fig 3D), suggesting that rear legs participate in body-wide leg coordination, despite their specializations. To visualize these patterns, we calculated the percentage of video frames during which a particular leg was simultaneously in swing phase with each other legs, thereby defining the most common 'swing partners' (Fig 5A). For

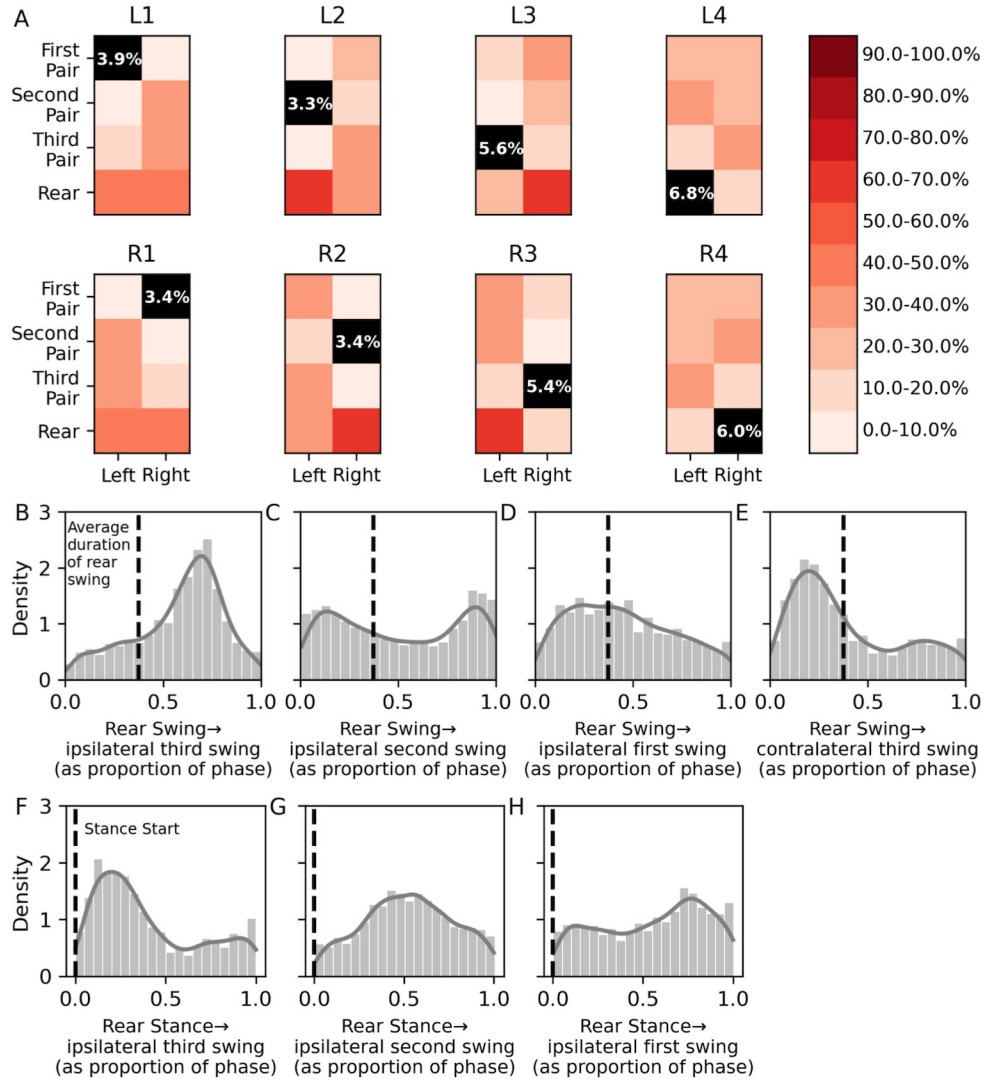

**Fig 5. Rear legs participate in body-wide coordination.** A. For legs indicated at the top of the grids, colors indicate the percentage of video frames that each other leg is also swinging. Black squares = percentage of all frames during which that leg is the only leg swinging. B-E. Timing between rear leg swing duration (dashed lines) and swing initiation in ipsilateral third (B), second (C), first (D), and contralateral third (E) legs, normalized to the rear leg step period. (F-H). Timing between rear leg stance onset (dashed lines) and swing initiation of ipsilateral third (F), second (G), and first (H) legs, normalized to the rear leg step period.

lateral legs, swing partners were diagonally oriented, in canonical tetrapod and tripod patterns (S2A Fig). Swing partnerships between rear legs and lateral legs were more common than lateral leg-lateral leg partnerships, presumably because rear legs spend more time in swing phase. Notably, swing partnerships between third legs and diagonally-oriented rear legs were particularly strong, whereas swing partnerships between rear legs and ipsilateral third legs were weaker (Fig 5A). These partnerships suggest that despite the unique characteristics of rear legs, coordination between rear legs and third pair legs is similar to the coordination within lateral legs.

If this coordination occurs, swings of third legs should be suppressed during swings of ipsilateral rear legs, and this suppression should be released at stance onset of the ipsilateral rear

leg. We compared rear leg swing and stance timing with the timing of swing initiation in all ipsilateral legs (Fig 5B–5H). Swings of third legs were tightly coupled to the swing durations and stance onsets of ipsilateral rear legs (Fig 5B and 5F). Moreover, rear legs and diagonally-oriented third pair legs tended to swing simultaneously (Fig 5E). These data suggest that rear legs participated in body-wide coordination, despite their distinct step kinematic parameters and anatomical specialization.

Within lateral legs, the swing initiation of the first leg pair is slightly delayed compared to the second leg pair (S4G Fig). This observation, combined with the slightly larger duty factor for anteriormost legs (Fig 3C), suggests a 'leading role' for forelegs in locomotion, as suggested by observations in *Drosophila* [19]. Furthermore, some leg swings were initiated before stance onset of neighboring posterior legs (S4F and S4G Fig); in insects, these early swings typically occur at highest speeds or shortest step periods [30].

## Preferred walking speeds and stepping patterns

Many animals have distinct ICPs at different speeds, for example, from walking to trotting to galloping in quadrupeds [2, 11, 35]. In these animals, transitions between gaits are marked by abrupt shifts in phase offsets and/or duty factors. In contrast, panarthopods like *Drosophila* and tardigrades show gradual changes in duty factor and phase offsets as speed changes [5, 17, 22]. For example, slow-moving *Drosophila* adults primarily utilize a pentapod ICP, and shift to tetrapod and tripod patterns with increased speed [5, 19, 22]. In addition, insects show higher variation in ICPs at low speeds [22, 36].

To determine if tardigrades have speed-dependent preferences for certain ICPs, we combined step timing with per-frame body locations, and plotted the speed of each lateral leg stride (5175 strides in 103 tardigrades, Fig 6A). From discontinuities within this distribution, we partitioned strides into low speed (n = 1550, < 0.23 body lengths s$^{-1}$), medium speed (n = 1992, between 0.23 and 0.35 body lengths s$^{-1}$), and high speed (n = 839, > 0.35 body lengths s$^{-1}$), and plotted phase offsets in each partition (Fig 6B). At low speeds, $\phi_I$ showed peaks at 1/3 and 2/3. For a tetrapod ICP, $\phi_I < 0.5$ generates an anterograde—from posterior to anterior, or meta-chronal [30] —wave of swings, while $\phi_I > 0.5$ generates an apparent retrograde wave (S2B and S2C Fig), which occurs when the step period is shorter than the complete metachronal cycle [30]. Thus slow-moving tardigrades exhibited both anterograde ($\phi_I = 1/3$) and apparent retro-grade ($\phi_I = 2/3$) waves of leg swings within a tetrapod ICP (Fig 6C). With increased speed, $\phi_I$ converged at 1/3, generating a canonical tetrapod gait with an anterograde wave of swings (Fig 6C). Distribution of $\phi_C$ was broad at low speeds, but converged around 1/2 as speed increased (Fig 6B). These changing distributions of $\phi_I$ and $\phi_C$ suggest that at high speeds, a canonical tet-rapod gait with asynchronous stepping is preferred, whereas a canonical tetrapod gait with synchronous stepping is also present at low speeds.

To determine how ICP composition changes with speed, we quantified the percentage of frames spent in each hexapod (Fig 6D) and rear (Fig 6E) ICP within each speed partition. At low speeds, a pentapod pattern was the predominant hexapod configuration, followed by canonical tetrapod. With increased speed, pentapod patterns decreased, while tetrapod and tri-pod patterns increased (Fig 6D). For rear legs, steps and hops became more prevalent as speed increased, whereas the stand pattern decreased (Fig 6E).

## Stepping patterns, walking speeds, and substrate characteristics

We employed a substrate of constant stiffness, enabling testing for correlation between speed and interleg coordination, independent of substrate composition. In contrast, Nirody and colleagues [17] compared tardigrade walking between stiff and soft substrates. They reported that

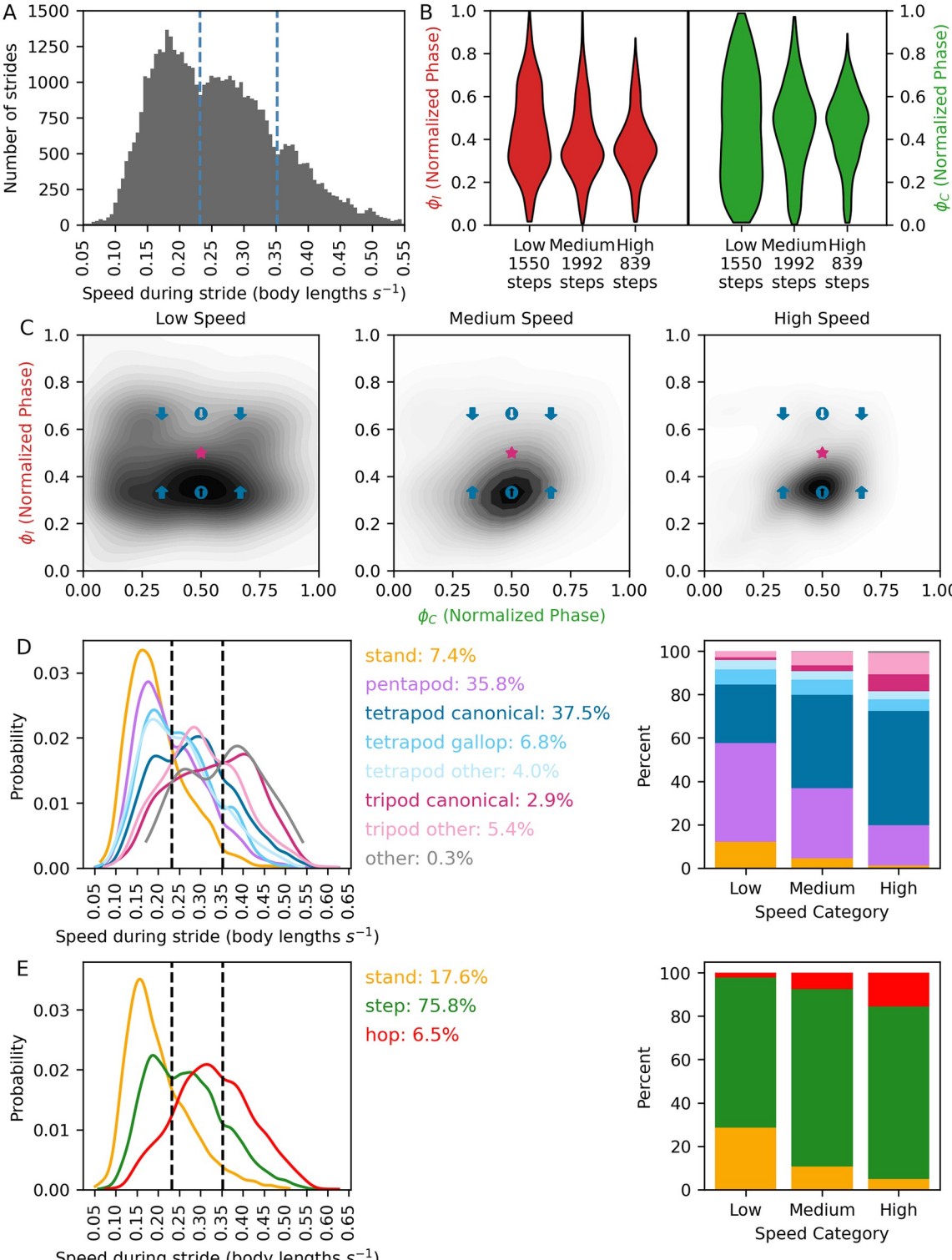

**Fig 6. Tardigrade preferences in walking speed and ICPs.** A. Distribution of walking speed (body lengths s⁻¹) for 5175 lateral leg steps. Walking speeds were subdivided into low, medium, and high partitions based on discontinuities in the distribution (dashed lines). B. Ipsilateral ($\phi_I$) and contralateral ($\phi_C$) phase offsets within speed partitions. C. Probability density plots of $\phi_C$ against $\phi_I$ in speed partitions, along with predicted stepping patterns at intersections of $\phi_C$ and $\phi_I$ as depicted in Fig 4E. D,E. Composition of hexapod ICPs for lateral legs (D), and rear ICPs (E). Left: Probability histograms of each individual stepping pattern at different speeds. Vertical lines = speed partition boundaries in (A). Center: percentage of all video frames wherein each particular ICP is observed. Right: Changes in composition of each ICP with increasing speed.

on soft substrates, tardigrades occasionally walked with a 'galloping' gait, had a bimodal distribution of $\phi_I$ with peaks at 1/3 and 2/3, and walked more slowly. These differences in interleg coordination may therefore depend on substrate stiffness, speed, or a combination of both. To quantify 'galloping' gait, where legs within a segment swing simultaneously, we subdivided tetrapod patterns into canonical tetrapod (S2A Fig), tetrapod gallop, and other (any other combination of two swinging legs). While the canonical tetrapod ICP predominated, most tardigrades also employed a galloping pattern (Figs 4G and 5D). Furthermore, slow-moving tardigrades in our dataset had a bimodal $\phi_I$ with peaks at 1/3 and 2/3 (Fig 6B), reminiscent of the tardigrades walking on a soft substrate in the study of Nirody et al. [17]. In their analysis, the prevalence of $\phi_I = 2/3$ in tardigrades walking on soft substrates was suggested to arise after an occasional galloping step wherein two legs within a segment swung simultaneously. After this gallop, one anterior leg would delay swinging and reset the tetrapod pattern (i.e. one anterior leg would have $\phi_I = 1/3$, while the paired anterior leg would have $\phi_I = 2/3$). If this resetting behavior is prominent, most strides with $\phi_I = 2/3$ (a delay to reset to the tetrapod pattern) would be followed by strides where $\phi_I = 1/3$ (maintaining the tetrapod pattern). We tested this prediction by collecting strides in third legs where $\phi_I$ was close to 2/3 (n = 277), plotted $\phi_I$ for the subsequent stride of that leg, and found evidence of sustained walking with $\phi_I = 2/3$, rather than resets to $\phi_I = 1/3$ (Fig 4K). In addition, sustained strides at $\phi_I = 2/3$ would produce a normalized metachronal lag of 4/3. In our dataset, the distribution of metachronal lag showed a primary peak at the canonical tetrapod value of 2/3 (2 strides along one side at $\phi_I = 1/3$), and a shoulder at 4/3, again suggesting some sustained walking at $\phi_I = 2/3$ (Fig 4L). Because $\phi_I = 2/3$ was sustained in our dataset, especially at slower speeds, tardigrades occasionally walk for extended periods with an apparent retrograde wave of swings (Fig 6C). At faster speeds, the distribution of $\phi_I$ converged to 1/3, while the distribution of $\phi_C$ converged to 1/2, similar to fast-moving tardigrades on stiff substrates [17]. These differences in interleg coordination may therefore arise from differences in intrinsic walking speeds, rather than compensatory changes in stepping patterns due to substrate characteristics.

## Comparing locomotion between juveniles and adults

Our dataset provides a rich source of information about intra-species variation in exploratory locomotion, step kinematics, and interleg coordination (S1 Table), which can be used as a baseline for comparisons across life stages, tardigrade species, and experimental treatments. We quantified more than 50 parameters of locomotion for each tardigrade, including measurements of exploratory locomotion, step kinematics, and interleg coordination (S1 Table). Many of these parameters are intercorrelated. To compare locomotion between treatments, we selected a subset of 21 measurements based on independence (e.g. include duty factor but exclude stance duration) or particular interest (e.g. include both absolute speed and body-length normalized speed). We ran pairwise comparisons for these measurements, and corrected for multiple comparisons at FDR < 0.05 via the Benjamini-Yekutieli procedure (Fig 7A–7C).

To compare locomotion between life stages, we isolated 21 juveniles within 24 hours of hatching, and compared them with our adult dataset (n = 103) (S2 Movie, S2 Table). While adult tardigrades were not precisely age matched, they were all much larger than juveniles. Adults were $\geq$ 192 μm in length (192–326 μm, mean 251 μm), while juveniles ranged from 124 to 150 μm (mean 137 μm, Fig 7A). Juveniles moved at slower absolute speeds, but speeds were equivalent to adults when normalized to body length (Fig 7A). These differences were reflected in step kinematics: juveniles took smaller lateral leg steps in absolute distance (Fig 7D), but larger steps when normalized to body length (Fig 7E). In addition, stance and swing duration

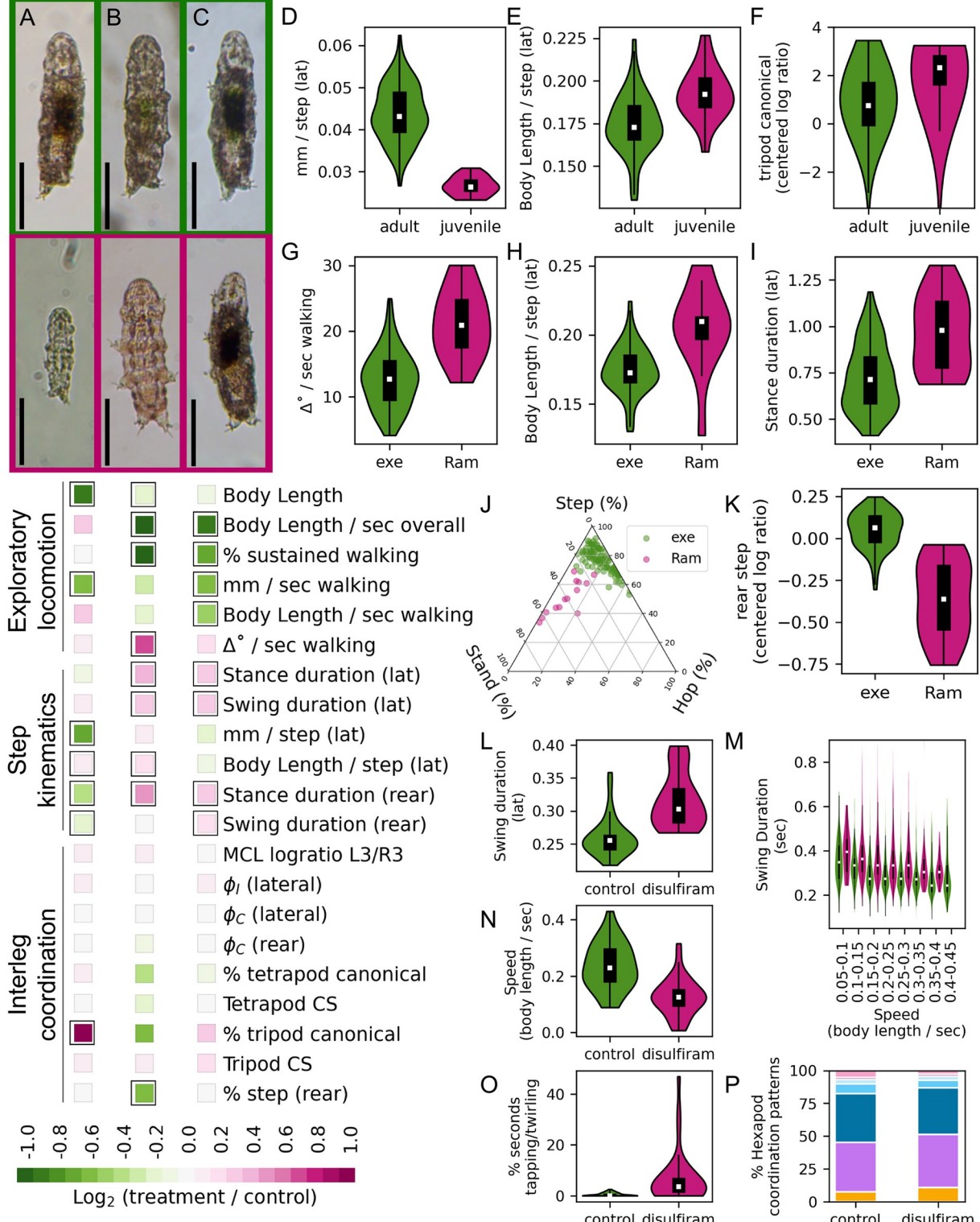

**Fig 7. Comparisons of walking across lifespan, species, and disulfiram treatment.** A-C. Representative images (top) and comparisons of locomotion (bottom) between the indicated pairs. Scale bars = 100 μm. A. Adults (n = 103) vs. Juveniles (n = 21). B. *H. exemplaris* (exe, n = 103) vs. *Ramazzottius* (Ram, n = 24). C. Control (n = 33) vs. disulfiram-treatment (n = 34). Colored squares show Log₂ (treatment / control) for each parameter, and black boxes indicate significance at FDR < 5% after correction for multiple comparisons. D. Juveniles take smaller steps in absolute distance compared to adults (p < 0.001). E. Juveniles take larger steps compared to adults, when steps are normalized to body size

(p < 0.001). F. Juveniles spend more time in a canonical tripod gait, compared to adults (p = 0.01). Compositional data underlying the percentage of time spent in canonical tripod gait were logratio transformed prior to comparison. G. *Ramazzottius* individuals turn more during each second of walking (p < 0.001). H. *Ramazzottius* individuals take longer steps with respect to body length (p < 0.001) I. *Ramazzottius* individuals have longer stance durations (p < 0.001). J. Ternary plot for compositional data of rear stepping patterns. K. Rear legs of *H. exemplaris* spend more time in a stepping gait compared to *Ramazzottius* (p < 0.001). Compositional data for rear stepping patterns were logratio transformed prior to comparison. L. Disulfiram-treated tardigrades exhibited longer swing durations (p < 0.001). M. Swing duration plotted against partitioned speed; swing durations of disulfiram-treated tardigrades were greater than controls for all speed partitions. N. Disulfiram-treated tardigrades moved more slowly (p < 0.001). O. Disulfiram-treated tardigrades exhibited more leg tapping and twirling than controls (p < 0.001). P. Comparison of percentage of video frames spent in each hexapod ICP (control vs. disulfiram-treated tardigrades), following the color scheme in Fig 2B.

of rear legs was shorter in juveniles, indicating a decrease in step period and thus a higher turnover rate (Fig 7A). Phase offsets during sustained walking were indistinguishable between adults and juveniles. Compared to adults, however, juveniles spent more time in a canonical tripod ICP (Fig 7F). Thus locomotion and coordination were similar between juveniles and adults, but juveniles spent more time in an ICP associated with higher speeds.

## Comparing locomotion between species

To analyze locomotor differences between species, we compared *Hypsibius exemplaris* (n = 103) to wild-caught tardigrades (S2 Movie, S2 Table). Captured tardigrades were grouped according to morphological characteristics (e.g. claw morphology and body pigmentation), and 24 were assigned to the *Ramazzottius* genus and selected for comparison. We sequenced a portion of 18S rDNA from nine of these individuals, and found that the top GenBank matches for all sequences were from *Ramazzottius*.

*Ramazzottius* individuals moved slower, and spent less time in sustained walking (Fig 7B, S2 Table). Speeds during walking bouts were similar, but direction of motion was more variable in *Ramazzottius* (Fig 7B and 7G). Accordingly, while stance and swing durations were longer in *Ramazzottius*, these individuals traveled further per step (Fig 7H and 7I). Remarkably, phase offsets and coordination patterns of lateral legs were indistinguishable between species, whereas the rear legs of *Ramazzottius* exhibited less time in 'step' mode and more time in 'stand' mode (Fig 7J and 7K). Thus while exploratory locomotion was quite different, lateral leg coordination during walking bouts was largely indistinguishable between species.

## Disulfiram treatment and locomotion

To disrupt locomotion via drug treatment, we selected disulfiram for its potential to broadly impact nervous system function (S2 Movie, S2 Table). Disulfiram inhibits a variety of metal-containing enzymes, including acetaldehyde dehydrogenase [37], dopamine ß-hydroxylase [38], and tyramine ß-hydroxylase [39]. Accordingly, disulfiram has been employed to inhibit octopamine synthesis in *Drosophila* [25]. *Drosophila* mutants with disrupted octopamine synthesis show reduced locomotor performance, including flight deficiencies and decreased walking speed [22, 40]. In humans, chronic disulfiram exposure has been associated with peripheral neuropathy and ataxia [26].

Disulfiram-treated tardigrades (n = 34) were much slower than controls (n = 33) overall, and during bouts of sustained walking (Fig 7C and 7N, S2 Table). Accordingly, stance and swing durations were longer in disulfiram-treated individuals (Fig 7C and 7L). During swing phase, disulfiram-treated tardigrades occasionally exhibited unusual movements: legs twirled or tapped before stance onset (Fig 7O and S3 Movie). To determine if increased swing durations after disulfiram treatment were due to decreased speed or from other differences during swing phase, we partitioned steps according to speed, and compared swing duration in each

speed partition. Across all speeds, swing durations were longer in disulfiram-treated tardigrades (Fig 7M). Thus disulfiram-treated tardigrades show decreased walking speed, increased stance and swing phase duration (and increased stride period), and unusual behaviors during swing phase. Even with these differences, phase offset timing and ICPs were indistinguishable from controls after disulfiram treatment (Fig 7C and 7P).

## Discussion

### Quantification of tardigrade locomotor behavior

Our study provides a comprehensive view of variation in tardigrade locomotor behavior within and between species, across life stages, and between experimental treatments. First, we developed tools for automated tracking of exploratory locomotion, and analyzed these behaviors in 103 adult *Hypsibius exemplaris* individuals. Per-frame changes in speed and direction of travel were quantified as tardigrades roamed over large distances (average of 22 body lengths) and times (average of >90 seconds). Second, we identified bouts of sustained walking in these videos and measured step kinematic data for an average of >16 strides for each of the 8 legs, over an average of >14 seconds. Third, we quantified interleg coordination via several complementary approaches during sustained forward walking. These analyses extend a previous study of tardigrade locomotion [17], where exploratory behavior was not tracked, and step kinematic and interleg coordination data were analyzed for an average of 8.3 seconds (vs. 14.4 in ours) for 43 tardigrades on divided between two substrates (vs. 103 in ours on a single substrate). Our accessible, automated setup facilitates throughput, enabling a thorough exploration of intraspecies variation and rapid screening for differences across treatments. Mechanisms of differences in exploratory locomotion detected in automated initial screens can be elucidated by quantifying step kinematics and interleg coordination within the same videos.

Planar walking, as analyzed here and previously [17, 41], is likely uncommon in natural habitats of tardigrades. Our unconstrained system is compatible with future studies that include three-dimensional substrate features. Thus our system could be employed to analyze behaviors typical of more natural habitats, including klinokinetic behaviors, wherein animals vary rates of turning, depending on environmental features [41]. A disadvantage of our setup is that measurements of footfall geometry, including step positioning, leg angles, and individual leg speeds [19, 20] are difficult in an unconstrained environment.

### Intraspecies variation in step kinematics and interleg coordination

Our study greatly expands the dataset of analyzed tardigrade strides. Nirody and colleagues [17] recorded 23 tardigrades on high-stiffness polyacrylamide gels, and 20 on gels of low-stiffness. For the high-stiffness substrate, the primary focus of their analysis, they analyzed 989 lateral leg strides and 382 rear leg strides, or ~60 complete strides per tardigrade. After optimization, we chose a single substrate (2% agarose), and analyzed locomotion in 103 tardigrades, with a total of 9471 lateral leg steps and 3692 rear leg steps, or ~128 complete strides per tardigrade.

For time-dependent step kinematic parameters (stance and swing duration, duty factor, step period), our measurements were between the high stiffness and low stiffness values found previously [17], suggesting that tardigrades walk similarly on agarose and acrylamide. However, for distance-dependent parameters (body length, distance per step), our measurements were about 50–70% of those found in the previous study. The source of these discrepancies is not clear, but the tardigrade size measurements in that study (mean length = 350 μm) were significantly larger than previous measurements of body length in *Hypsibius exemplaris*

(mean = 232 µm, range = 112–293 µm, n = 30) [18], and the values we report here
(mean = 251.5 µm, range = 192–326 µm, n = 103). We verified tardigrade size via independent
measurements (Fig 1A), and our values are consistent with several other studies that report
body size in this species [13, 42]. In addition, we verified the taxonomic identity of tardigrades
by 18S rDNA sequencing.

Measurements of interleg coordination were consistent with the previous study [17]. Tardigrades employed a canonical tetrapod coordination pattern 38% of the time (vs. 31% previously), and 3% in a canonical tripod pattern (vs. 3%). In addition, we quantify a larger range of
coordination patterns, including non canonical patterns like 'galloping', by assigning a
momentary ICP to each video frame [20, 21]. Moreover, we confirm that these momentary
snapshots of canonical tetrapod or tripod patterns lead to sustained patterns of coordination
through time, by calculating the coordination strength (CSS) during each bout [21, 22], and by
calculating a coordination consistency score (CCS) for each tardigrade.

## Speed-dependent preferences in coordination patterns

Panarthropods such as insects and tardigrades transition smoothly between stepping patterns
as speed changes, and can employ several ICPs at a given speed [1, 17, 19]. Although transitions between ICPs in these animals are smooth, there is evidence that walking speed distributions are not continuous. Rather, certain speeds are preferred, and are associated with
characteristic stepping patterns. For example, walking speed in cockroaches clustered into two
groups [43], and specific walking speeds were underrepresented in a study of *Drosophila* gait
[19].

From discontinuities in the distribution of tardigrade walking speeds, we subdivided strides
into low, medium, and high speeds. At low speeds, the distribution of $\phi_I$ was bimodal, with a
prominent peak at $\phi_I = 1/3$ and a secondary peak of $\phi_I = 2/3$, while the distribution of $\phi_C$ was
broad (Fig 6B). Intersections of $\phi_I$ and $\phi_C$ at 1/3 and 2/3 were prominent at low speeds, indicating tetrapod gait with synchronous stepping and a mixture of anterograde and apparent retrograde waves of leg swings (Fig 6E and S2B and S2C Fig). Our observations of tardigrade
stepping at slow speeds were similar to tardigrades walking on soft substrates in the previous
study [17], suggesting that this pattern of coordination depends primarily on walking speed,
rather than compensatory changes in coordination caused by differences in substrate stiffness.

As tardigrade speed increased, $\phi_I$ converged to 1/3, and $\phi_C$ converged to 1/2, consistent
with a tetrapod gait with asynchronous, anterograde waves of leg swings. Though multiple
ICPs were present in each speed partition, representation of different ICPs changed with
increased speed (Fig 6D and 6E). Pentapod patterns prevailed at slower speeds, while canonical
tetrapod and tripod patterns became more prominent at faster speeds. Thus our dataset provides evidence for preferred walking speeds, with a mixture of ICPs within each speed, and
speed-dependent preferences in step patterns (e.g. tetrapod with synchronous stepping at slow
speeds, tetrapod with asynchronous stepping at fast speeds). Preferences for speed and step
patterns may reflect optimal energetic economy or stability, or perhaps distinct neural and
motor programs at different speeds [19].

## Specialized rear legs participate in body-wide coordination

Interestingly, despite distinct aspects of function, anatomy, and step kinematics of rear legs,
their step period is similar to that of lateral legs, especially at higher speeds (Fig 3D). This similarity suggests that rear legs may participate in body-wide interleg coordination. Ipsilateral
fourth and third legs apparently followed the coordination rules that govern relationships
between ipsilateral lateral legs [31]; third leg swings were suppressed when the ipsilateral rear

leg was swinging, and this suppression was released upon stance onset of the rear leg (Fig 5B and 5C). Accordingly, rear legs tended to swing simultaneously with contralateral third legs and ipsilateral second legs, creating patterns similar to canonical tetrapod and tripod ICPs (Fig 5A and 5E). Thus rear legs have distinct step kinematic parameters, yet are intercoordinated with the rest of the legs.

## Newly-hatched juveniles walk like adults

After embryonic stages, there is little increase in cell number in tardigrades, though some specialized cells do proliferate [42]. Juveniles are thus self-sufficient, miniature versions of adults [44]. Accordingly, parameters of juvenile and adult walking were largely identical, and juveniles moved as rapidly as adults when speed was normalized to body length (Fig 7A, 7D and 7E). However, juveniles appeared to spend more time in a tripod ICP, which is associated with faster speeds (Figs 5D and 7F). This observation matches previous studies in stick insects, where first instar nymphs primarily employed a tripod gait, while adults mostly walked in a tetrapod pattern [45]. Graham [45] hypothesized that this difference was related to the size of the body relative to objects in the environment. Larger animals adjust their steps to navigate obstacles, and may require the additional stability from four legs contacting the ground. In contrast, smaller animals navigate around or over these obstacles.

## Similar interleg coordination despite different exploratory behavior between species

Compared to *Hypsibius exemplaris*, *Ramazzottius* individuals were slower, walked less, turned more, and had longer stance and swing durations (Fig 7B, 7G and 7K). During sustained walking, the timing of interleg phase relationships was similar between species, as was time spent in canonical tetrapod and tripod stepping ICPs. Thus while high-level aspects of locomotion (speed, turns, stops, and starts) differed, interleg coordination during walking bouts was conserved. These results are similar to those observed by Nirody and colleagues [17], in which phase timing relationships were similar despite significantly different speeds on substrates of different stiffnesses. The differences between H. exemplaris and Ramazzottius could potentially arise from effects of long-term culturing in the lab; further work will be required to investigate this possibility.

## Disulfiram disrupts exploratory behavior but not interleg coordination

Disulfiram-treated tardigrades were slower, had fewer bouts of sustained walking, and had longer stance and swing durations. Despite these differences, interleg coordination was unaffected in drug-treated tardigrades (Fig 7C and 7P). While the mechanism by which disulfiram disrupts locomotion in tardigrades remains unknown, one possibility is that disulfiram disrupts octopamine synthesis by inhibiting tyramine ß-hydroxylase [25, 39]. Importantly, tardigrades have a predicted 1:1 orthologous relationship (GenBank accession number OQV12655) via reciprocal BLAST with the *Drosophila* tyramine ß-hydroxylase protein sequence. *Drosophila* mutants lacking octopamine synthesis have deficiencies in locomotor performance [22, 40], and octopamine modulates walking initiation in cockroaches [46]. Interestingly, octopamine appears to selectively regulate high-level aspects of locomotion in insects. For example, *Drosophila* lacking octopamine show deficiencies in flight initiation and maintenance, but have normal wingbeat frequencies and amplitudes during flying bouts [40]. Similarly, *Drosophila* lacking octopamine walk slower and less frequently, yet have normal step kinematics and interleg coordination during walking bouts [22]. Behaviors of disulfiram-treated tardigrades were consistent with these results, in which absence of octopamine caused defects in high-level

control of movement (e.g. initiation and maintenance of locomotion), yet interleg coordination was unaffected, suggesting that the processes that regulate these two aspects of movement are distinct.

Alternatively, the tapping and twirling motions observed in disulfiram-treated tardigrades, along with speed-independent increases in swing duration (Fig 7M), could indicate disrupted sensory feedback or proprioception. *Drosophila* adults with inactivated leg sensory neurons walk slower but maintain interleg coordination [19]. In these sensory-deficient *Drosophila*, step precision and leg function was disrupted, leading to uncoordinated body and foot placement. These defects were especially apparent in slow-moving individuals; flies walking at fast speeds were less dependent on sensory feedback, suggesting movement at different speeds employs distinct locomotor mechanisms [19]. As in *Drosophila*, regulation of interleg coordination in tardigrades is apparently decoupled from neuronal programs that control initiation and maintenance of walking.

Taken together, comparisons across species and treatments suggest that neuronal control of high-level aspects of walking (e.g. speed, stops, starts, maintenance, and turns) is independent from circuits governing step kinematics and interleg coordination.

### Tardigrades as a model for locomotion and movement disorders

Locomotion requires interactions between regions of the nervous system, nerves and muscles, and organisms and their environment. Computational analysis of locomotion facilitates integration of these distinct aspects of biology [4]. Even with a relatively simple nervous system, patterns of limb coordination are conserved in tardigrades. The jointless motion of tardigrade legs may enable a more straightforward model of neuronal control of limb-based locomotion, compared to systems like insects where individual joints are controlled separately [1]. Importantly, genes underlying many human movement disorders are conserved in tardigrades. *Hypsibius exemplaris* individuals are transparent, which will facilitate functional studies using reporter genes, calcium indicators, and optogenetic analyses, and genetic toolkits are actively being developed [16]. These tools, combined with the relatively small numbers of tardigrade neurons, may help unlock the potential for cellular-level analyses of appendicular locomotion.

### Supporting information

**S1 Fig. Body size vs. speed.** A-B. Walking speed is proportional to body length (A, Spearman's $\rho = 0.371$, $p < 0.001$, $n = 103$) and body area (B, Spearman's $\rho = 0.343$, $p < 0.001$, $n = 103$). C. When normalized to body length, walking speed is not correlated with body size. Least squares regression fit lines and 95% confidence intervals were calculated in the Python package seaborn.
(TIF)

**S2 Fig.** Interleg coordination patterns A. Examples of interleg coordination patterns for lateral legs and rear legs. B-E. Phase offsets determine hexapod stepping patterns. Duty factor for lateral legs is set at 2/3. B. When $\phi_I = 1/3$ and $\phi_C = 1/3$ or $2/3$, a canonical tetrapod pattern is produced, with leg pairs initiating swing phase simultaneously (synchronous), and an anterograde (or metachronal) wave of swing initiations that travels from back to front. C. When $\phi_I = 2/3$ and $\phi_C = 1/3$ or $2/3$, a canonical tetrapod pattern is produced, with leg pairs initiating swing phase simultaneously (synchronous), and an apparent retrograde wave of swing initiations that travels from front to back. D. When $\phi_I = 1/3$ or $2/3$ and $\phi_C = 1/2$, a canonical tetrapod pattern is produced, with leg pairs initiating swing phase asynchronously. When $\phi_I = 1/3$, the wave of leg swings is anterograde, while when $\phi_I = 2/3$, the swing initiation pattern is

apparently reversed. E. When $\phi_I = 1/2$ and $\phi_C = 1/2$, a canonical tripod gait is produced.
(TIF)

**S3 Fig. Phase offsets in individual lateral legs.** A. Distribution of ipsilateral phase offsets ($\phi_i$) for each individual second or third pair leg. Timing of swing initiation between the indicated leg and the neighboring ipsilateral anterior leg, expressed as a fraction of the period of the posterior leg. n (strides) = 1301 for L2, 1279 for L3, 1317 for R2, 1278 for R3. B. Distribution of contralateral phase offsets ($\phi_C$) for each left lateral leg. Timing of swing initiation between the indicated leg and the right leg within the same segment, expressed as a fraction of the period of the left leg. Probability density estimates of the distributions were calculated via the Python package seaborn. n (strides) = 1305 steps for L1, 1251 for L2, 1253 for L3.
(TIF)

**S4 Fig. Suppression and initiation of neighboring leg swings.** A. Cumulative distribution functions of timing of anterior ipsilateral and within-segment contralateral leg swings, normalized to the phase of the posterior or contralateral lateral leg. Average swing duration of lateral legs is shown by the dashed line, n = 5175 strides for ipsilateral phase offsets; n = 3809 strides for contralateral offsets. B. Ipsilateral swing timing for each lateral leg in the first and second leg pair, n (strides) = 1301 for L2, 1279 for R2, 1317 for L3, 1278 for R3. C. Contralateral swing timing for each left lateral leg, n (strides) = 1305 for L1, 1251 for L2, 1253 for L3. D. Cumulative distribution function of timing of contralateral leg swings for rear legs. Average swing duration of rear legs is shown by the dashed line. E. Contralateral swing timing for each rear leg, n (strides) = 1530 for L4, 1532 for R4. F. Cumulative distribution functions of timing of anterior ipsilateral and within-segment contralateral leg swings, normalized to the phase of the posterior or contralateral lateral leg. Stance onset of lateral legs is shown by the dashed line, n = 4571 strides for ipsilateral stance-swing offsets; n = 3548 strides for contralateral stance-swing offsets. G. Ipsilateral swing timing for each lateral leg in the first and second leg pair, compared to stance onset of the reference leg, n (strides) = 1145 for L2, 1154 for R2, 1145 for L3, 1127 for R3. H. Contralateral swing timing for after stance initiation of left lateral leg, n (strides) = 1203 for L1, 1184 for L2, 1161 for L3.
(TIF)

**S1 Table. Measurements of exploratory locomotion, step kinematics, and interleg coordination.**
(DOCX)

**S2 Table. Comparison of tardigrade locomotion across life stage, species, and disulfiram treatment.**
(DOCX)

**S1 Movie. Tracking of tardigrade size, speed, and direction.**
(MP4)

**S2 Movie. Comparisons of tardigrade locomotion.**
(MP4)

**S3 Movie. Leg taps and twirls after disulfiram treatment.**
(MP4)

## Acknowledgments

We thank the IC Tardigrade Tracks classes of 2021–2023 for methods development, drug treatment optimization, and capture of wild tardigrades, and Brooks Miner for discussions and advice.

## Author Contributions

**Conceptualization:** Te-Wen Lo, Ian G. Woods.

**Data curation:** Ian G. Woods.

**Formal analysis:** Ian G. Woods.

**Funding acquisition:** Te-Wen Lo, Ian G. Woods.

**Investigation:** Emma M. Anderson, Sierra G. Houck, Claire L. Conklin, Katrina L. Tucci, Joseph D. Rodas, Kate E. Mori, Loriann J. Armstrong, Virginia B. Illingworth, Te-Wen Lo, Ian G. Woods.

**Methodology:** Emma M. Anderson, Sierra G. Houck, Claire L. Conklin, Katrina L. Tucci, Joseph D. Rodas, Kate E. Mori, Loriann J. Armstrong, Virginia B. Illingworth, Te-Wen Lo, Ian G. Woods.

**Project administration:** Te-Wen Lo, Ian G. Woods.

**Resources:** Ian G. Woods.

**Software:** Ian G. Woods.

**Supervision:** Ian G. Woods.

**Validation:** Ian G. Woods.

**Visualization:** Ian G. Woods.

**Writing – original draft:** Te-Wen Lo, Ian G. Woods.

**Writing – review & editing:** Joseph D. Rodas, Virginia B. Illingworth, Te-Wen Lo, Ian G. Woods.

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
