## [Decision Letter · Decision Letter 0]

29 Aug 2024

PONE-D-24-33965Comparative analysis of tardigrade locomotion across life stage, species, and pharmacological treatmentPLOS ONE

Dear Dr. Woods,

Thank you for submitting your manuscript to PLOS ONE. After careful consideration, we feel that it has merit but does not fully meet PLOS ONE’s publication criteria as it currently stands. Therefore, we invite you to submit a revised version of the manuscript that addresses the points raised during the review process.

We look forward to receiving your revised manuscript.

Kind regards,

Bob Goldstein

Academic Editor

PLOS ONE

Journal Requirements:

"National Science Foundation Louis Stokes Alliances for Minority Participation (award number 2110082)

William T. Grant Foundation (Grant ID #ODF-203161)

Ithaca College Collegiate Science and Technology Entry Program

Ithaca College Humanities and Sciences Summer Scholars Program"

4. Please note that funding information should not appear in the Acknowledgments section or other areas of your manuscript. We will only publish funding information present in the Funding Statement section of the online submission form. Please remove any funding-related text from the manuscript. 

5. Please note that your Data Availability Statement is currently missing the repository name (http://www.github.com/enwudz/gai;
http://doi.org/10.17605/OSF.IO/XBKA2) or a direct link to access each database (GenBank database). If your manuscript is accepted for publication, you will be asked to provide these details on a very short timeline. We therefore suggest that you provide this information now, though we will not hold up the peer review process if you are unable.

**Additional Editor Comments:**

You will see that both reviewers are enthusiastic about your manuscript, although both have minor concerns that will need to be addressed in your revision.

Reviewers' comments:

Reviewer's Responses to Questions

**Comments to the Author**

1. Is the manuscript technically sound, and do the data support the conclusions?

Reviewer #1: Yes

Reviewer #2: Yes

2. Has the statistical analysis been performed appropriately and rigorously? 

Reviewer #1: Yes

Reviewer #2: Yes

3. Have the authors made all data underlying the findings in their manuscript fully available?

Reviewer #1: Yes

Reviewer #2: Yes

4. Is the manuscript presented in an intelligible fashion and written in standard English?

Reviewer #1: Yes

Reviewer #2: Yes

5. Review Comments to the Author

Reviewer #1: In this paper, Anderson et al. characterize tardigrade kinematics and coordination patterns using a large dataset that comprises two different life stages (juvenile and adult), two different species, and a drug treatment. Tardigrades are a fascinating organismal system for biomechanics studies, and it's exciting to see them being represented in the literature. This study is well constructed and the manuscript is clearly written. At this point, I only have a few comments, below:

- It seems that steps were pooled across individuals for analysis. Were tests done to ensure the independence of these measurements (e.g., comparing variances between animals). For instance, these were done in order to pool leg data (Figure 3).

- Are the arrows over the density plots (e.g., Figures 4E and 5C) providing useful information? I would suggest removing them as they are somewhat distracting and obscure the data below them.

Reviewer #2: The manuscript by Anderson et al. extends a previous study (by another group) that characterized kinematics and interleg coordination of tardigrades. Tardigrades are among the smallest animals with appendages, and the authors propose that the relative simplicity of the tardigrade body plan would facilitate studies to better understand the different factors that contribute to animal locomotion. To this end, the authors developed a series of scripts to automate analysis of tardigrade locomotion. Using these scripts, they compare locomotion in Hypsibius exemplaris adults to the previous study's data, using similar nomenclature and calculations to describe patterns of locomotion. The authors then extend these findings across longer walk times in H. exemplaris adults, present evidence that locomotion patterns are generally consistent across developmental time, and compare locomotion in H. exemplaris to a wild-caught population of Ramazzottius sp. Their data corroborate data from the previous study and further suggest that at least some features of locomotion (such as interleg coordination) are conserved among tardigrades.

The development of open source tools for automated analysis of tardigrade locomotion would allow for greater insight into how locomotion is coordinated within a relatively simple animal system, and such studies could have implications for understanding more complex interactions in other animals. The similarity between the data presented here and in Ref. 17 suggests that the automated tools are reliable, and it was interesting to see that the comparison of juvenile and adult tardigrade locomotion are in line with what has been observed in stick insects. However, some of the conclusions (especially related to the disulfiram experiment) were overstated.

Major Issues:

1. A key outcome of this study is the development of automated tools to analyze locomotion in tardigrades. While the Python scripts were made available on GitHub, the current organization and file naming system in the GitHub repository makes it difficult to determine which scripts were used for each calculation. It would be helpful to name each script when describing specific calculations and data within the manuscript (such as in the Step Kinematics section of the Methods, lines 101-109, p. 4). It would also be helpful to provide more details on how scripts were optimized and validated. For example, when describing automated measurements of body size and location (lines 203-209, p. 7), how were the measurements verified as part of testing the scripts? Given similarities between this study and that described in Ref. 17, it may be necessary to explain what distinguishes these Python scripts from those in Ref. 17 (since at least some scripts appear to have similar purposes).

2. The descriptions of some calculations could be made more clear, such as those for CSS and CCS.

3. In the Results (lines 344-346, p. 11), the absolute value of the log2 ratio for metachronal lag was reported as 0.34 (0.34 ± 0.25). However, in Table S1, the "Abs Metachronal lag Log2 Ratio" was listed as 0.47 ± 0.2. Is this a reporting error, or am I misinterpreting the calculation?

4. Regarding the disulfiram experiments, since disulfiram has not been used routinely in tardigrades, it would be helpful to learn more about how the exposure conditions were determined. Were the concentration and time of exposure used in this study determined empirically or were they based on other studies? Were the authors able to rule out more general effects of the drug (such as cytotoxicity) in correlating observed phenotypes with potential roles in nervous system function?

In addition to the experimental parameters, I also felt that the abstract and introduction implied that multiple pharmacological drugs would be examined in this study, whereas only a single comparison (disulfiram vs control) was actually included.

5. The authors note that the planar locomotion analyzed in this study and ref. 17 and 39 are likely unnatural for most tardigrade populations, I am curious if differences in "natural" habitats could contribute to the differences observed between H. exemplaris and Ramazzottius. The H. exemplaris cultures have been bred in lab environments for many generations, while the Ramazzottius population in this study were wild-caught. It would not be easy to address this issue unless it were possible to analye wild H. exemplaris, but it may factor into interspecies comparisons.

Minor Issues:

1. In-text citations switch from numerical format to Author-Year format at several points in the manuscript. Specifically, Author-Year format is used in lines 267 and 272 (p. 9) and lines 400 - 408 (pp. 12 - 13).

2. Line 598, p.18: "sustain" should be "sustained"

3. Figure 4 is out of sequence, appearing after Figure 7 in the manuscript.

4. Figure 7A-C: The figure legend for Figure 7 does not identify the specific pairings shown in panels A-C (although they are identified in the text of the results). Specifying the pairings in the figure legend would improve clarity.

5. Figure 7P: I am assuming that the chart is following the same color scheme as shown in Figure 2B? It would be helpful to note this as part of the figure legend.

6. PLOS authors have the option to publish the peer review history of their article (what does this mean?). If published, this will include your full peer review and any attached files.

Reviewer #1: No

Reviewer #2: No

---

## [Author Response · Author response to Decision Letter 0]

2 Sep 2024

[These were also uploaded in the document: "Response to Reviewers.docx"

Dear editors,

Thanks very much for your comments on our manuscript, and for forwarding the comments of the two reviewers. We are excited about the prospect of our work being published in PLOS ONE.

Point-by-point responses to editor and and reviewer comments are below, discussing the changes we made to the manuscript in response to these comments. 

In addition to these suggested changes, we updated the legend of Figure 7 to add the p-values for the paired comparisons shown; we inadvertently omitted these from the submitted manuscript. We also made a few minor adjustments to the text to improve readability. These changes, along with the suggested changes from reviewer and editor comments, are visible in the 'Revised Manuscript with Tracked Changes' document.

Editorial Comments

1. If applicable, we recommend that you deposit your laboratory protocols in protocols.io to enhance the reproducibility of your results. Protocols.io assigns your protocol its own identifier (DOI) so that it can be cited independently in the future.

Detailed protocols for tardigrade handling, preparation for videography, video acquisition, path tracking, step timing quantification, and analysis have been posted at protocols.io. In addition, these posted protocols include descriptions of and instructions for the dPython scripts used in the analysis pipeline. These protocols are publicly available at http://dx.doi.org/10.17504/protocols.io.kxygxy8nwl8j/v1, and this link has been added to the manuscript under the 'Data Availability' heading. 

The manuscript is formatted according to the PLOS ONE style requirements.

We have ensured that these numbers match.

Our original cover letter has been amended (and reuploaded) to include this statement, and it is also included here in the rebuttal letter, just in case: "The funders had no role in study design, data collection and analysis, decision to publish, or preparation of the manuscript."

5. Please note that funding information should not appear in the Acknowledgments section or other areas of your manuscript. We will only publish funding information present in the Funding Statement section of the online submission form. Please remove any funding-related text from the manuscript. 

There is no funding information in the Acknowledgments or elsewhere in the manuscript.

6. Please note that your Data Availability Statement is currently missing the repository name (http://www.github.com/enwudz/gait;
http://doi.org/10.17605/OSF.IO/XBKA2) or a direct link to access each database (GenBank database). If your manuscript is accepted for publication, you will be asked to provide these details on a very short timeline. We therefore suggest that you provide this information now, though we will not hold up the peer review process if you are unable.

Repository names and direct links to databases have been added to the Data Availability Statement, including the new addition of publicly available protocols at protocols.io.

Captions for Supporting Information have been added to the end of the manuscript, and in-text citations have been updated to match.

The reference list was checked for completeness and correctness; no retracted papers were cited.

We uploaded our figures to PACE prior to the original submission, and have also done so for the figures in the resubmitted manuscript. 

Reviewer Comments

Reviewer One

1. It seems that steps were pooled across individuals for analysis. Were tests done to ensure the independence of these measurements (e.g., comparing variances between animals). For instance, these were done in order to pool leg data (Figure 3).

We thank the reviewer for their thoughtful comments and suggestions. 

In the original version of the manuscript posted to bioRxiv, we pooled steps in the plot of step kinematic data presented in Figure 3, to enable a direct comparison with the Nirody et al study (ref 17), in which step data was similarly pooled. However, in the manuscript submitted to PLOS One, steps were not pooled for Figure 3; instead, we now show the within-individual averages for step kinematic parameters. The plot characteristics and major conclusions were identical in both approaches.

For the distributions of phase offsets in Figure 4, S3 Figure, S4 Figure, and Figure 6, we pooled steps to enable direct comparisons to results presented in Nirody et al. (ref 17), who similarly plotted offsets from pooled step data. We confirm their results and add new observations about preferences for ICPs at different step speeds, and about coordination between rear legs and lateral legs. Averaging the ipsilateral offset values per individual would obscure the fact that these offsets are bimodally distributed (Fig 4A), consisting of peaks at 1/3 and 2/3, both of which lead to canonical tetrapod stepping patterns.

In the bottom of Figure 4, we pooled data for metachronal lag and ipsilateral offsets in successive steps to directly test whether individual steps with ϕI = 2/3 were subsequently 'corrected' to the canonical value of ϕI = 1/3, or if extended bouts of steps at ϕI = 2/3 were observed in the dataset. In this case, it is the extended bouts of steps at ϕI = 2/3 that are interesting, rather than variation or comparisons in these bouts between individuals. Like the ipsilateral offsets, the distribution of metachronal lag has separate peaks that arise from combinations of ϕI = 1/3 and ϕI = 2/3 in successive steps, and per-individual averaging would obscure this underlying structure.

Steps were pooled across individuals for Figure 5 (in the submitted manuscript, now Figure 6 in the revised manuscript) to provide a large sample to show discontinuities in the distribution of normalized speed during each step. Similar discontinuities were observed when we plotted the average step speeds per individual (in 06_Speed_Partitions.ipynb from the GitHub repository, www.github.com/enwudz/gait), as shown below:

[IMAGE AVAILABLE IN UPLOADED DOCUMENT]

For all other analyses and comparisons (e.g. between treatments in Fig 7), the data compared are within-individual averages.

2. Are the arrows over the density plots (e.g., Figures 4E and 5C) providing useful information? I would suggest removing them as they are somewhat distracting and obscure the data below them.

These arrows show how specific intersections of ipsilateral and contralateral phase offset values produce canonical tetrapod and tripod coordination patterns (S2 Fig). In the resubmitted manuscript, we have greatly decreased the size of these arrows in Fig 4E and Fig 5C (now Fig 6C) to better showcase the data beneath (i.e. prominent intersections of ipsilateral and contralateral phase offsets colocalize with canonical tetrapod and tripod coordination patterns, as depicted in Fig 4E). Note that Fig 5C in the submitted manuscript is now Fig 6C, as we reordered the figures to match the text (thanks to a comment by Reviewer Two, below).

Reviewer Two

1. A key outcome of this study is the development of automated tools to analyze locomotion in tardigrades. While the Python scripts were made available on GitHub, the current organization and file naming system in the GitHub repository makes it difficult to determine which scripts were used for each calculation. It would be helpful to name each script when describing specific calculations and data within the manuscript (such as in the Step Kinematics section of the Methods, lines 101-109, p. 4). It would also be helpful to provide more details on how scripts were optimized and validated. For example, when describing automated measurements of body size and location (lines 203-209, p. 7), how were the measurements verified as part of testing the scripts? Given similarities between this study and that described in Ref. 17, it may be necessary to explain what distinguishes these Python scripts from those in Ref. 17 (since at least some scripts appear to have similar purposes).

Thanks very much for these suggestions. In response, we have taken several additional steps to clarify the steps and scripts used in data acquisition and analysis, and have added text to the manuscript to direct readers to these resources (lines 107-108). First, all code to produce the tables and figures are included in the GitHub repository as Jupyter notebooks, and each notebook is named according to the figure or table they produce. In addition to code, these notebooks contain instructions for making each figure. Second, detailed protocols for video acquisition and step timing quantification, including how and when to use each script in the GitHub repository, have been posted publicly to protocols.io (http://dx.doi.org/10.17504/protocols.io.kxygxy8nwl8j/v1). Because multiple scripts are employed in several steps (e.g. three scripts are used in the recording, analysis, and visualization of step kinematic data), we felt that inclusion of all script names in the manuscript would add clutter and perhaps confusion. 

Optimization and validation: We have provided additional details in the manuscript (lines 97-99, 104-107) about how the automated measurements and tracking were validated. We validated size measurements in several ways. First, measurements of size were taken in every video frame, providing internal controls for the automated detection of tardigrade outlines. Second, we validated the automated measurements for several tardigrades via manual measurement in additional image analysis programs (e.g. ImageJ, Photoshop, Preview), and via independent scripts that we developed (e.g. manualCritterMeasurement.py and measureImage.py in our GitHub repository). Third, our measurements of adult H. exemplaris size are consistent with published values for this species, whereas average sizes reported by Nirody et al. are larger than these published values.

We validated the automated tracking in four ways. First, for most clips we followed the tracking 'live' as the computer identified and plotted the position, orientation, and speed of the tardigrade during each video clip. Second, we plotted the tracked path for all video clips (e.g. as in Figure 1D) and reran the tracking (e.g. with adjustments to sensitivity or specificity parameters) for any clips where the plotted path raised questions about tracking accuracy. Third, we used the tracking data to make videos showing the walking tardigrade along with the real-time tracked path, stops, and turns for a subset of our video clips (e.g. as in S1 Movie). Fourth, the step tracking scripts use the tracking data to crop and rotate each video frame to facilitate tracking of each leg. Any errors in tracking would be evident in aberrantly cropped or rotated frames in these analyses.

Distinguishing scripts from those in Ref 17. There are three major features of our scripts that distinguish them from those in the previous publication. First, our method of video acquisition enabled tardigrades to roam freely over the surface of the agarose, allowing us to perform automated tracking of exploratory behavior, which is a key feature of our study. In the previous study, tardigrades were manually tracked as they walked across a stationary field of view. Our tardigrades were thus tracked for much longer, and data was combined over several clips for each tardigrade. Second, we collected data that was not obtained in the previous study (e.g. direction of motion, starts and stops). These differences in data acquisition led to a difference in data formatting, which necessitated development of novel analysis scripts. Third, we measured or calculated many parameters that were not collected in the previous study, including quantification of coordination strength and consistency, symmetry of step kinematic parameters across the body, metachronal lag, and change in direction per step. These new measurements are calculated at various points in the analysis pipeline, and necessitated the development of new scripts.

2. The descriptions of some calculations could be made more clear, such as those for CSS and CCS.

We have added text to clarify CCS (lines 145-152) and CSS (lines 157-161). In addition, the CCS calculation method is illustrated in Fig 2H.

3. In the Results (lines 344-346, p. 11), the absolute value of the log2 ratio for metachronal lag was reported as 0.34 (0.34 ± 0.25). However, in Table S1, the "Abs Metachronal lag Log2 Ratio" was listed as 0.47 ± 0.2. Is this a reporting error, or am I misinterpreting the calculation?

Thank you for the very careful review. The correct value is shown in the S1 Table. The results text (lines 383-384) has been updated with this correct value.

4. Regarding the disulfiram experiments, since disulfiram has not been used routinely in tardigrades, it would be helpful to learn more about how the exposure conditions were determined. Were the concentration and time of exposure used in this study determined empirically or were they based on other studies? Were the authors able to rule out more general effects of the drug (such as cytotoxicity) in correlating observed phenotypes with potential roles in nervous system function? In addition to the experimental parameters, I also felt that the abstract and introduction implied that multiple pharmacological drugs would be examined in this study, whereas only a single comparison (disulfiram vs control) was actually included.

We added text (lines 192-194) to the manuscript to clarify how we determined the effective dosage, and we changed the title, abstract, and Fig 7 title to focus specifically on disulfiram. Initial dose ranges were taken from exposure estimates based on previous studies in Drosophila, and from long-term use in humans. The dose used in this study was 

---

## [Editor Report · Decision Letter 1]

6 Sep 2024

Comparative analysis of tardigrade locomotion across life stage, species, and disulfiram treatment

PONE-D-24-33965R1

Dear Dr. Woods,

We’re pleased to inform you that your manuscript has been judged scientifically suitable for publication and will be formally accepted for publication once it meets all outstanding technical requirements.

Kind regards,

Bob Goldstein

Academic Editor

PLOS ONE

---

## [Editor Report · Acceptance letter]

10 Sep 2024

PONE-D-24-33965R1 

PLOS ONE

Dear Dr. Woods, 

I'm pleased to inform you that your manuscript has been deemed suitable for publication in PLOS ONE. Congratulations! Your manuscript is now being handed over to our production team.

Kind regards, 

on behalf of

Dr. Bob Goldstein 

Academic Editor

PLOS ONE